# Climatic Chamber Stability Tests of Lipase-Catalytic Octyl-Sepharose Systems

Tomasz Siódmiak [1,*], Joanna Siódmiak [2], Rafał Mastalerz [3], Natalia Kocot [3], Jacek Dulęba [3], Gudmundur G. Haraldsson [4], Dorota Wątróbska-Świetlikowska [1] and Michał Piotr Marszałł [3]

1. Department of Pharmaceutical Technology, Faculty of Pharmacy, Medical Biotechnology and Laboratory Medicine, Pomeranian Medical University in Szczecin, 71-251 Szczecin, Poland
2. Department of Laboratory Medicine, Faculty of Pharmacy, Ludwik Rydygier Collegium Medicum, Nicolaus Copernicus University in Toruń, 85-094 Bydgoszcz, Poland
3. Department of Medicinal Chemistry, Faculty of Pharmacy, Collegium Medicum in Bydgoszcz, Nicolaus Copernicus University in Toruń, 85-089 Bydgoszcz, Poland
4. Science Institute, University of Iceland, 107 Reykjavik, Iceland
* Correspondence: tomasz.siodmiak@pum.edu.pl

**Abstract:** The application of the climatic chamber presented in this paper to assess the storage stability of immobilized lipases is a new approach characterized by the potential of unifying the study conditions of biocatalysts created in various laboratories. The data achieved from storing lipases in the climatic chambers may be crucial for the chemical and pharmaceutical industry. Our paper describes the developed protocols for immobilization via interfacial activation of lipase B from *Candida antarctica* (CALB) and lipase OF from *Candida rugosa* (CRL-OF) on the Octyl-Sepharose CL-4B support. Optimization included buffers with different pH values of 4–9 and a wide range of ionic strength from 5 mM to 700 mM. It has been shown that the optimal medium for the CALB immobilization process on the tested support is a citrate buffer at pH 4 and high ionic strength of 500 mM. Implementing new optimal procedures enabled the hyperactivation of immobilized CALB (recovery activity 116.10 ± 1.70%) under the applicable reaction conditions using olive oil as a substrate. Importantly, CALB storage stability tests performed in a climatic chamber under drastic temperature and humidity conditions proved good stability of the developed biocatalyst (residual activity 218 ± 7.3% of dry form, after 7 days). At the same time, the low storage stability of CRL OF in a climatic chamber was demonstrated. It should be emphasized that the use of a climatic chamber to test the storage stability of a dry form of the studied lipases immobilized on Octyl-Sepharose CL-4B is, to our knowledge, described for the first time in the literature.

**Keywords:** lipase B from *Candida antarctica*; lipase from *Candida rugosa*; stability tests; Octyl-Sepharose; immobilization protocols

## 1. Introduction

The direction of biocatalysis development is inseparably correlated with the "green chemistry" conception [1]. Reduction of waste production and the application of renewable resources are particularly important aspects of this subject [2–4]. This trend is useful in chemical synthesis due to, among others, the selection of non-toxic reagents and mild reaction conditions. The implementation of enzymatic catalysts significantly limits the negative impact on the environment. Moreover, it reduces reaction costs. The utilization of enzymes in catalysis allows for decreasing the use of organic solvents and shortens the reaction time, due to fewer reaction steps. For this reason, biocatalysts are commonly applied in industrial processes [5].

Lipases are among the most commonly used catalysts in obtaining compounds with pharmaceutical importance [6–8]. A significant proportion of enzymes from this class contain a specific polypeptide chain (known as the "lid"), covering their active site [9]. The



lid is characterized by a changeable length and amphipathic character [10–12]. The shift of the lid, conditioned by the medium in which the lipase exists, determines the movement of equilibrium to open or closed lipase conformation [13]. In closed form, the hydrophilic part of the lid is directed toward the reaction medium, whereas the hydrophobic part interacts with the hydrophobic area of the active site. On the other hand, in open (active) conformation, the huge hydrophobic pocket exposes the active site to the reaction medium. This phenomenon, commonly named interfacial activation, occurs when, in the presence of an oil drop, the lid moves upon contact with a hydrophobic surface, and the lipase open form is adsorbed onto the hydrophobic area of the drop. A change of equilibrium towards the open form transpires and the action at the water/oil interface starts [14–16]. One of the most widely applied lipases in the pharmaceutical industry and commonly described in science reports is the lipase B from *Candida antarctica* (CALB) and a lipase derived from *Candida rugosa* (CRL) [17,18].

Lipase B from *Candida antarctica* belongs to the $\alpha/\beta$-hydrolases family, with a catalytic triad containing Ser-Asp-His. CALB possesses two mobile $\alpha$-helices ($\alpha5$ and $\alpha10$), surrounding the active center, which can act as a lipase lid and contributes to the ability of the enzyme to interact with many different substrates. It should be mentioned that the mechanism of the catalytic activity of the CALB remains the subject of research, to answer the issue of whether CALB catalyzes its reactions by interfacial activation [16,19]. The optimum pH of the reaction medium for CALB is 7.4. It is worth observing that the enzyme activity decreases in the medium with pH below 6 and above 8. This phenomenon is probably related to the ionization state of the amino acid residues of Asp 187 and His 224 from the catalytic triad [20]. The isoelectric point of CALB is at pH 6 [21]. The described lipase is also characterized by high enantioselectivity, which is reflected in its widespread use in the pharmaceutical industry for the preparation of, e.g., pure drug enantiomers [22–25].

The lipase from *Candida rugosa* is a protein with a molecular weight of 60 kDa, belonging to, similarly to CALB, the $\alpha/\beta$-hydrolases family. The CRL is characterized by high catalytic activity, low costs, and diverse substrate specificity. Thanks to the polypeptide chain, the "lid", in the presence of a hydrophobic surface, CRL undergoes interfacial activation, which allows the hydrolysis of poorly soluble substrates in water (oils and fats), unlike standard esterases [26–28]. Importantly, yeast *Candia rugosa* is especially powerful in secreting a subset of five lipase isoforms (identical in 77%), which are characterized by the ability to hydrolyze lipids and esters of cholesterol [29–31]. The CRL specificity is conditioned by the molecular properties of the enzyme, substrate structure, and factors affecting enzyme–substrate interactions [30].

The application of lipase in industry and academic studies needs a search for new technological solutions to increase enzyme catalytic activity and stability. The modification of lipase properties by immobilization is especially noteworthy. Immobilization is a process based on, among others, physical adsorption, entrapment, cross-linking, as well as a covalent binding between the functional group of the support and the enzyme [32–35]. This method should, as mentioned above, increase the catalytic activity and stability of lipase. Furthermore, it can facilitate the separation of the catalyst from the reaction medium, which results in the possibility of reusing the enzyme, thus reducing reaction costs. As was described by Bolivar et al. [36], the immobilization system should include crucial aspects, such as the appropriate immobilization protocol (support activation, enzyme immobilization conditions, enzyme–support interactions), proper support, and a suitable active group in the support. In the immobilization design stage, it should consider enzyme stability, the tune of selectivity or specificity, and resistance to inhibition as well as enzyme purification coupled to enzyme immobilization. The development of immobilization conditions includes the evaluation of the influence of, e.g., temperature, pH, reaction time, and the reaction medium. The popular technique is lipase immobilization onto hydrophobic supports, where interfacial activation occurs during the adsorption process [37–39]. As mentioned above, an extremely important aspect of immobilization optimization is the proper selection of the support. Materials, functioning as supports, used in the immobiliza-

tion should be characterized by suitable porosity, surface, hydrophilicity/hydrophobicity properties, and the presence of defined reactive groups. Moreover, the low cost of production, least possible environmental pollution, resistance to the influence of chemical and physical factors, reusability and increased specificity, non-toxicity, and surface preventing microbial growth, etc., are also the issues necessary to be considered [40–44]. Hence, the selection of optimal support is a crucial part of the catalytic system design.

The supports used in immobilization are usually of non-organic and organic origin. Amongst the wide group of non-organic supports, especially applied are, inter alia: silica, aluminum, ceramics, glasses, and magnetic nanoparticles. These supports are microbiologically neutral and have beneficial mechanical properties. In turn, the matrix of organic supports consists mainly of polyacrylamide derivatives, polyvinyl alcohol, polyamides, or polysaccharides (e.g., starch, cellulose, chitosan, agarose) [45]. The particular research interest is directed toward a linear heteropolysaccharide—agarose. It is characterized by high hydrophilicity, mechanical resistance, varied size of particles and pores, and the presence of a significant amount of hydroxyl groups, which allow the modification of its surface by cross-linking or the attachment of suitable functional groups. The agarose can be modified to create such derivatives as, among others, glyoxyl-octyl agarose, alkyl-aldehyde agarose, and octyl-glutamic agarose, which give the possibility of, e.g., multipoint attachment, and increase the activity and stability of immobilized lipases. An example of commercially applied support, due to the possibility of lipase immobilization in open form, is modified agarose—Octyl-Sepharose CL-4B (octyl-agarose) [46,47].

One of the most important features of enzymatic catalytic systems, apart from high enzymatic activity, is their high stability to various factors such as temperature, pH of the medium, addition of ions, or organic solvents. The high stability of obtained catalytic models is of extreme desire for their application in the chemical and pharmaceutical industries. Numerous authors have performed stability tests [47,48].

When analyzing the literature data, one should pay attention to the lack of uniform guidelines for testing the stability of catalytic systems. For this reason, due to the wide application of immobilized models in biocatalysis, an attempt was made to standardize enzyme stability tests and develop equal standards. Therefore, the climatic chamber was introduced to assess, under well-controlled and homogeneous conditions (in accordance with pharmaceutical standards for drugs), the effect of temperature, humidity, and light (in the Vis range) on the stability of the developed catalytic systems. This experiment, as an attempt to set a standard for testing the stability of lipase catalytic models, has not been previously mentioned in the literature.

In the presented publication, the optimization of immobilization of the lipase B from *Candida antarctica* and the lipase from *Candida rugosa* was described. Accordingly, the effect of pH buffer and ionic strength have been tested on the CALB and CRL-OF. The storage stability studies have been performed in a climatic chamber for both immobilized lipases, in dry form. The analysis was performed using the titrimetric method.

## 2. Results and Discussion

### 2.1. Lipase CALB Immobilization Protocols

#### 2.1.1. Effect of Buffer pH

CALB was immobilized on the Octyl-Sepharose CL-4B support via interfacial activation using buffers with different pH values. Lipolytic activity was tested, and the results were presented as the activity recovery parameter. The obtained data are shown in the graph in Figure 1. The amount of the lipase immobilized on the support was also determined using the Bradford method.

The results of the assessment of the buffer pH value impact used for the immobilization process show the lowest value of lipase activity recovery after immobilization in Trizma Base buffer at pH 9 (49.90 ± 1.42%). The highest value of the determined parameter was achieved for CALB immobilized in citrate buffer at pH 4 (116.10 ± 1.70%), i.e., below the lipase isoelectric point, which is at pH 6.0 [49]. By analyzing the obtained values, we

can observe a significant influence of the pH of the buffer used for the immobilization process on the catalytic activity of the immobilized CALB (as the pH value increases, the enzyme activity decreases). This fact may be related to the effect of the buffers on the net charge of lipase or the ionization state of the catalytic triad [50], as well as the effect of the buffers on the dissociation state of the CALB and the interaction between the enzyme and the support [51]. Therefore, optimizing this parameter when developing optimal catalytic systems is necessary. These results are consistent with the studies by Brigida et al. [52]. The authors carried out the CALB immobilization process by adsorption of the enzyme and received the highest value of lipase activity recovery using a pH 4 buffer for the immobilization process. The activity recovery parameter was defined as the ratio of enzymatic activity of the immobilized enzyme and the total units of soluble lipase that disappeared from the supernatant during immobilization (the parameter of activity recovery presented in our paper is defined in Section 3.6). It was found that the activity recovery values depend on the immobilization pH because the interaction between the molecule and its medium influences the structure of the protein molecule. Therefore, it is believed that these interactions may contribute to an increase in the catalytic activity of enzymes. It is assumed that carrying out immobilization with the use of a citrate buffer of pH 4 may have a positive effect on the interaction of lipase with the medium. It is worth noticing that the pH value of the buffer used may also affect the enzyme molecules' tertiary structure and active site [53].

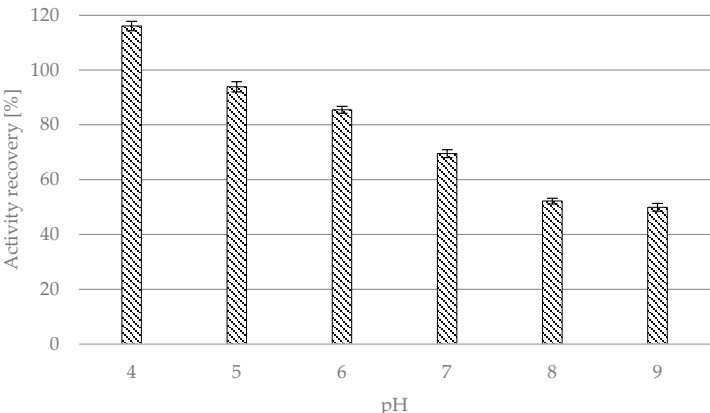

**Figure 1.** Effect of buffer pH on CALB activity recovery. Reaction conditions: CALB immobilized onto Octyl-Sepharose CL-4B (50.0 mg), or free lipase (10.0 mg), phosphate buffer (100 mM, pH 7.4), the emulsion of gum Arabic and olive oil, temperature 37 °C, incubation 30 min. Data are presented as means ± standard deviations of three analyses ($n$ = 3). The error bars represent the standard deviations of the mean.

Tanasković et al. [54] studied the effect of pH (in the range of 4 to 10) of buffers used to immobilize CALB on metakaolin support. The highest value of activity immobilization yield, in the hydrolysis reaction of *p*-nitrophenyl butyrate (*p*-NPB) as a substrate, was obtained using acetate buffer at pH 5 for immobilization. A decrease in the value of this parameter was observed with an increase in the pH value of the buffer used, similar to the results presented in our work. The pH 4 buffer showed lower catalytic activity than pH 5, which was explained by the unfavorable orientation of the enzyme during the immobilization process and the molten globule phenomenon, which can cause a decrease in activity.

Arana-Pena et al. [48] immobilized CALB on Octyl-Sepharose CL-4B support using phosphate buffer at pH 7 and a low ionic strength of 5 mM. The researchers perceived that the CALB activity slightly decreased during immobilization (to 80%). This result proves the reduced catalytic activity of the immobilized form of the enzyme compared to the free form. The authors explained this by the structure of CALB being characterized by a small lid that does not isolate the active center of the enzyme. In our study, using a pH 7 phosphate buffer,

activity recovery of 69.50 ± 1.45% was gained. It should be noted that the results described in our work indicate that the use of a citrate buffer with pH 4 allows for achieving higher activity recovery values than the process carried out using buffers with higher pH values. It is worth emphasizing that, apart from the pH value, the chemical compound used to prepare the buffers must be taken into account, as well as the substrate used in the reaction (olive oil in our project). Importantly, in our studies, the highest amount of catalytic protein immobilized on the support was also observed in the pH 4 acid buffer. The lipase loading values were as follows: in the pH 4 buffer, the lipase loading was 64.20 ± 1.40 mg/g, in the pH 7 buffer, 42.40 ± 1.81 mg/g, while in the pH 8 buffer, 36.20 ± 1.31 mg/g. On the other hand, the protein immobilization yield was as follows: in the pH 4 buffer, 32.10 ± 0.70%, in the pH 7 buffer, 21.20 ± 0.905%, while in the pH 8 buffer, 18.10 ± 0.655%. It is worth noting that the amount of immobilized catalytic protein decreased with the increased pH value of the buffer used. Similar trends for adsorbed protein were presented for CALB immobilized on green coconut fiber [52]. Thus, selecting the appropriate pH value of the buffer is one of the crucial steps in optimizing immobilization. Due to the highest activity recovery value, it was decided that the procedure of CALB immobilization in citrate buffer at pH 4 would be used for further tests.

### 2.1.2. Effect of Ionic Strength

CALB was immobilized with the application of buffers of pH 4 and different ionic strengths. The lipolytic activity (U) was then assayed, and the relative activity was calculated from the obtained results. The values are shown in the graph in Figure 2.

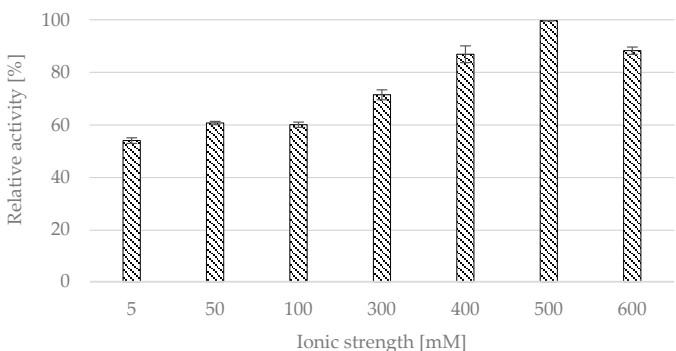

**Figure 2.** Effect of buffer ionic strength on the activity of immobilized CALB. Reaction conditions: immobilized CALB onto Octyl-Sepharose CL-4B (50 mg), phosphate buffer (100 mM, pH 7.4), the emulsion of gum Arabic and olive oil, temperature 37 °C, incubation 30 min. Data are presented as means ± standard deviations of three analyses (*n* = 3). The error bars represent the standard deviations of the mean.

The presented results indicate an apparent influence of the ionic strength of the tested citrate buffer at pH 4 on the lipolytic activity of the immobilized enzyme. Among the tested immobilization conditions, the lowest relative activity was observed for lipase immobilized in a buffer with an ionic strength of 5 mM (54.05 ± 0.01%). On the other hand, the highest relative activity value was obtained using a buffer with an ionic strength of 500 mM for immobilization. A decrease in the determined parameter was noted above the value of 500 mM. Based on the results, it can be assumed that the increase in the enzyme's catalytic activity under conditions of high ionic strength (up to 500 mM) may result from the favorable orientation of the enzyme during immobilization.

It should be mentioned that the studies were conducted using olive oil as a substrate to determine lipolytic activity. Arana-Peña et al. [55] found that the procedure of CALB immobilization on hydrophobic supports is a repeatable and reliable method, but the situation may change if the substrate is changed. It was also recognized that the increase in ionic strength during enzyme immobilization negatively affected enzyme activity with *p*-NPB (*p*-nitrophenyl butyrate) as a substrate. However, analyzing the results achieved in

our studies, it can be seen that there is a strong relationship between the increase in the lipolytic activity of the enzyme using olive oil and the increase in the ionic strength of the buffer used during immobilization. Cespugli et al. [56] immobilized CALB onto the rice husks and methacrylic resins using phosphate buffer with high ionic strength of 0.5 M at pH 8. Tributyrin was used as a substrate for assessing lipolytic activity. Another group, Mihailović et al. [57], immobilized CALB onto resin support using phosphate buffer (0.05 M, 0.5 M, or 1 M, pH 7.00). The hydrolytic activity of immobilized lipase was determined using *p*-nitrophenyl butyrate (*p*-NPB) as a substrate. The highest values of expressed activity were obtained using a buffer with a high ionic strength of 1 M for immobilization. They pointed out that hydrophobic interactions occur at higher ionic strengths, and these conditions often result in a more open conformation of the immobilized lipase. They noted the initial concentration of lipase used for immobilization being a fundamental issue. It should be mentioned that the literature also describes the conditions for the immobilization of lipases on the Octyl-Sepharose support with the use of neutral pH buffers (pH 7) and low ionic strength (5 mM) [48]. Analyzing the results presented in our work, it seems evident that the dependence of the effect of ionic strength on CALB activity requires further research and analysis, taking into account the structure of the lipase lid (its size or lack of it), the type of binding, supports, and substrates used. Additionally, Tanasković et al. [54] indicated that the buffer concentration significantly impacts the strength and nature of interactions between the enzyme and the support. They emphasized that high ionic strength conditions favor hydrophobic interactions between the enzyme and the support due to the exposure of hydrophobic regions of the protein molecule. In the literature, there are also different opinions regarding the effect of the high ionic strength of the buffer on the exposition of the large hydrophobic pocket of the enzyme [9].

The pH of the buffer may also influence the behavior of the lipase when using a high ionic strength. Studies of changes in the catalytic activity of lipases in citrate buffer at pH 4, at different ionic strengths, are described to a minimal extent in the literature and require a broader discussion. Our results indicate that the modification of the ionic strength of the buffer used for immobilization allows for significant changes in the catalytic parameters of the lipase under the tested reaction conditions. Due to the highest catalytic activity of lipase in the citrate buffer with pH 4 and an ionic strength of 500 mM, these conditions were used for further project steps. The CALB immobilization scheme is presented below (Figure 3).

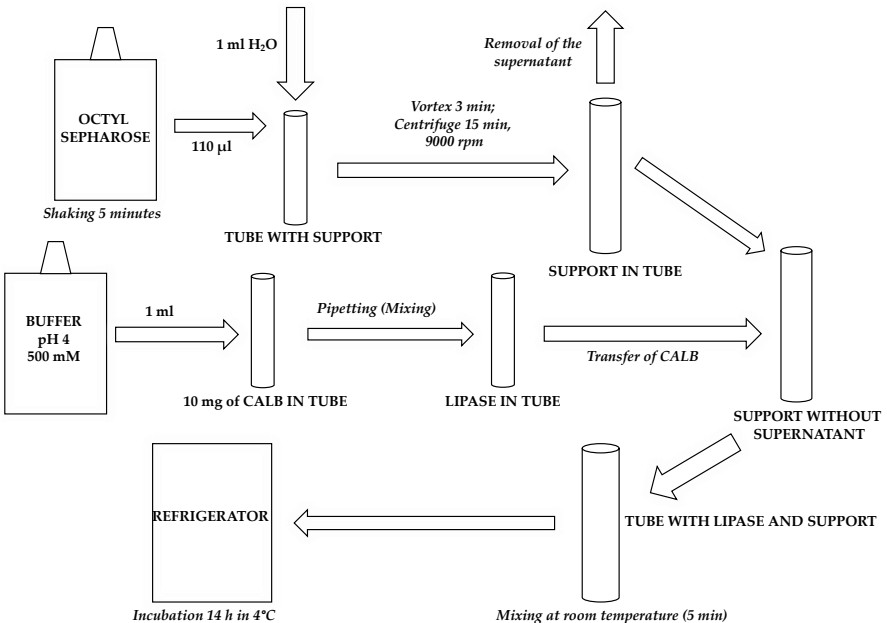

**Figure 3.** The scheme of the CALB immobilization onto Octyl-Sepharose CL-4B support.

### 2.2. Lipase from Candida Rugosa (CRL OF) Immobilization Protocols

The CRL-OF and CALB enzymes used in our research significantly differ in their structure. CRL-OF, unlike CALB, has a lid that fully separates the active center of the enzyme from the external medium, whereas the presence of a lid in CALB is still a matter of debate (Section 1.). It should be considered that the differences in the structure of the tested enzymes can determine the different catalytic activity of the immobilized CRL-OF obtained using the optimal immobilization protocol for CALB. Based on the results achieved in the optimization stage of CALB immobilization on the Octyl-Sepharose CL-4B support, critical process parameters were selected. Parameters such as pH and ionic strength of the buffer significantly affect the change in the catalytic activity of the lipase. Therefore, the process of optimizing the pH and ionic strength of the buffer used for CRL-OF immobilization on the Octyl-Sepharose CL-4B support was carried out. The tests were performed using the process duration and temperature conditions used in the CALB immobilization protocol.

#### 2.2.1. Effect of pH

CRL-OF was immobilized on Octyl-Sepharose CL-4B support via interfacial activation using buffers with an ionic strength of 100 mM and varying buffer pH values. Then, the determination of lipolytic activity (U) was carried out. Based on the obtained results, the activity recovery was calculated. The achieved values are presented in Table 1.

**Table 1.** Influence of buffer pH on the value of lipolytic activity (U) and activity recovery CRL-OF.

| pH | U (Immobilized CRL-OF) | Activity Recovery (%) |
|---|---|---|
| 4 | 34.37 ± 2.81 | 22.86 ± 0.02 |
| 7 | 37.65 ± 2.84 | 25.04 ± 0.02 |
| 9 | 10.78 ± 1.08 | 7.17 ± 0.01 |

Reaction conditions: immobilized CRL-OF onto Octyl-Sepharose CL-4B (50 mg), or free lipase (10 mg), phosphate buffer (100 mM, pH 7.4), the emulsion of gum Arabic and olive oil, temperature 37 °C, incubation 30 min. Data are presented as means ± standard deviations of three analyses ($n = 3$).

The immobilization procedure was carried out in buffers with pH values of 4, 7, and 9 to evaluate the effect of different pH values on the lipolytic activity of the lipase. Noteworthy is the low activity recovery values obtained for each tested buffer. The values of the determined parameter decreased along with the increase of the buffer pH value. The lowest activity recovery value was recorded for CRL-OF immobilized with a buffer of pH 9. It is assumed that the alkaline medium used for immobilization can negatively affect the conformation of the immobilized lipase and contribute to the inactivation of the enzyme by promoting structural changes in the catalytic protein, and through it, a drastic reduction in catalytic activity. Additionally, diffusion limitations may occur when testing lipolytic activity with olive oil. It is worth noting that the immobilization of CRL-OF in a buffer of pH 4 allowed us to receive an activity recovery value more than 3 times higher than using a buffer of pH 9 for immobilization. Similar trends in the behavior of CRL-OF in buffers of different pH values were described by Arana-Peña et al. [55]. The researchers immobilized the lipase from *Candida rugosa* in buffers at pH 5, 7, and 9. The immobilization process using buffers at pH 5 and 7 allowed receiving the results of *p*-NPB hydrolysis activity on a similar level, while a significant decrease in activity was detected using a pH 9 buffer for immobilization. Considering the results, a decision was made to use a pH 4 buffer for further studies, despite a slight difference in the activity recovery values obtained with a pH 7 buffer.

#### 2.2.2. Effect of Ionic Strength

CRL-OF was immobilized on Octyl-Sepharose CL-4B using buffers at pH 4 and various ionic strengths. Subsequently, the determination of lipolytic activity was performed. Based on the obtained results, the relative activity was calculated. The achieved values are presented on the graph in Figure 4.

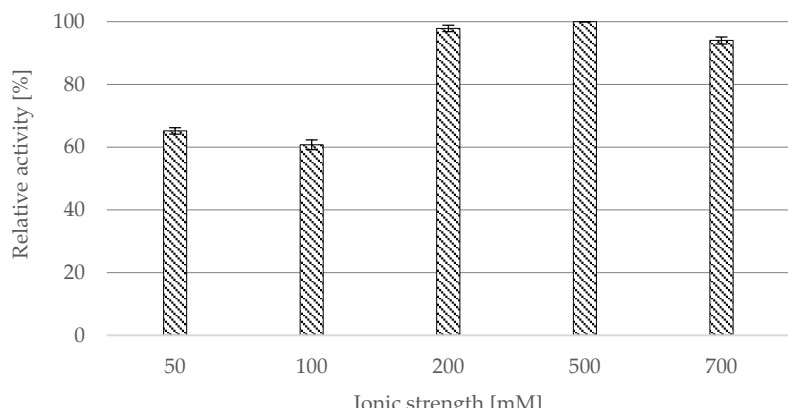

**Figure 4.** Effect of buffer ionic strength on CRL-OF relative activity. Reaction conditions: immobilized CRL-OF onto Octyl-Sepharose CL-4B (50 mg), phosphate buffer (100 mM, pH 7.4), the emulsion of gum Arabic and olive oil, temperature 37 °C, incubation 30 min. Data are presented as means ± standard deviations of three analyses (*n* = 3). The error bars represent the standard deviations of the mean.

The evaluation results of the lipolytic activity of immobilized CRL-OF present the highest relative activity value using immobilization in a buffer with a high ionic strength of 500 mM. It is worth noting that using a buffer with an ionic strength of 200 mM, a relative activity of 97.85 ± 0.01% was determined, while using a buffer with an ionic strength of 700 mM, 94.01 ± 0.01%. These results suggest a wide range of buffer ionic strengths at which CRL-OF shows the highest activity. However, as demonstrated in the pH effect studies, it should be taken into consideration that the lipolytic activity of the tested lipase is low (determined as activity recovery). It is worth noting that the CALB lipase indicated a more significant decrease in relative activity after immobilization in a 600 mM buffer than CRL-OF after immobilization in a 700 mM buffer, and there was a much narrower range of ionic strength at which CALB showed high activity. The lowest relative activity values were achieved when buffers with an ionic strength of 50 mM and 100 mM were used for the CRL-OF immobilization process.

In its structure, CRL-OF has a lid responsible for lipase adsorption on hydrophobic surfaces. It is assumed that this can affect the activity of the lipase in a wide range of ionic strength of the buffer. In the case of CALB, this range was much narrower, which may be due to the small polypeptide chain acting as the lid of this lipase. However, it should be noted that in both cases, the highest relative activity values were received in citrate buffer with an ionic strength of 500 mM. Thus, CALB, despite a smaller lid, behaves similarly to CRL-OF, presenting the maximum lipolytic activity at the same buffer value. It can be assumed that under high ionic strength conditions, the immobilized lipases exhibit a conformational equilibrium shifted towards the 'open' form of the enzyme. This structure arrangement determines the increase in the lipolytic activity of the enzyme compared to the closed conformation. Cea et al. [58] immobilized CRL onto biochar and evaluated, e.g., the effect of the increasing salt concentration (0–2 M KCl) in the solution media on the immobilization process. It was stated that low and high ionic strengths caused activity losses. It was indicated that a salt concentration higher than 0.25 M resulted in a decrease in activity, and residual activity of 54% was measured with 2.0 M KCl. De Melo et al. [59] described the immobilization of CRL on Octyl-Sepharose in a buffer with an ionic strength of 200 mM using *p*-nitrophenyl propionate (*p*-NPP) as a substrate, obtaining hyperactivation of the enzyme associated with interfacial activation, as a result of which the catalyst can be immobilized in its open form. They emphasized that the catalytic protein can be immobilized first by hydrophobic adsorption using a high ionic strength before forming the covalent bonds. Due to the received results, it was decided that for the next stage of research (stability tests), for CRL-OF, a pH 4 buffer with an ionic strength of 500 mM will be used.

### 2.3. Climatic Chamber Storage Stability Tests of CALB and CRL-OF

The storage stability studies of CALB and CRL-OF lipases in a dry form in a climatic chamber were carried out under drastic conditions of temperature (65 °C) and humidity (75%). In addition, the influence of light in the visible spectral range (400–800 nm) on the formed catalytic systems was investigated. After 7 days of storage in the climatic chamber, tests of lipolytic activity were performed and residual activity values were determined in relation to the value of the catalytic activity of lipases determined immediately after immobilization (without storage in the chamber). In addition, in order to compare the results of the CALB and CRL-OF stability studies from the climatic chamber, the immobilized lipases were also stored for 7 days in a refrigerator (4 °C). The obtained results are presented in Table 2.

**Table 2.** Effect of storage conditions in the climatic chamber on the storage stability of the immobilized CALB and CRL-OF, in dry form.

| Storage Conditions | | | Test Tube | CALB | | | CRL-OF | | |
|---|---|---|---|---|---|---|---|---|---|
| Temperature 65 °C | Humidity 75.0% | Vis 400–800 nm | Open/Closed/Dark Glass | CALB Residual Activity (%) | Activity (U) | Activity (U/g Support) | CRL-OF Residual Activity (%) | Activity (U) | Activity (U/g Support) |
| + | + | + | open | 171.3 ± 2.6 | 1.80 ± 0.13 | 36.04 ± 2.55 | 3.6 ± 0.6 | 1.30 ± 0.25 | 26.04 ± 5.09 |
| + | | + | closed | 218.8 ± 4.6 | 2.30 ± 0.17 | 46.04 ± 3.47 | 5.4 ± 0.6 | 1.97 ± 0.25 | 39.38 ± 5.09 |
| + | + | | dark glass open | 218.8 ± 7.3 | 2.30 ± 0.25 | 46.04 ± 5.09 | 4.0 ± 0.8 | 1.47 ± 0.25 | 29.38 ± 5.09 |
| + | | | dark glass closed | 203.0 ± 4.0 | 2.14 ± 0.21 | 42.71 ± 4.19 | 2.2 ± 0.2 | 0.80 ± 0.10 | 16.04 ± 1.92 |
| Refrigerator [4 °C] | | | closed | 203.0 ± 4.6 | 2.14 ± 0.17 | 42.71 ± 3.47 | 96.8 ± 5.2 | 35.47 ± 2.59 | 709.38 ± 51.75 |

Reaction conditions: immobilized CALB or CRL-OF onto Octyl-Sepharose CL-4B (50 mg), phosphate buffer (100 mM, pH 7.4), the emulsion of gum Arabic and olive oil, temperature 37 °C, incubation 30 min. Refrigerated storage conditions: temperature (4 °C), closed tube, no light. Data are presented as means ± standard deviations of three analyses (*n* = 3).

Analyzing the results obtained for CALB, it was observed that the immobilized lipase is characterized by excellent storage stability under the tested conditions. Moreover, it was noticed that storage in a climatic chamber and a refrigerator after immobilization, in a dry form, positively affected enzymatic activity. There was no significant effect of temperature (4 °C and 65 °C), humidity, and light on the value of the residual activity. The highest residual activity values were achieved for samples stored in the open dark and closed glass vials, 218.8 ± 7.3% and 218.8 ± 4.6%, respectively. It should be noted that the results of residual activity gained after storage in the climatic chamber and the refrigerator (at 4 °C) are higher than those received in the sample not subjected to the tested factors (determined immediately after immobilization). This may indicate a positive effect of optimal immobilization conditions: citrate buffer with pH 4 and ionic strength of 500 mM, on CALB storage stability. It is assumed that the immobilized CALB stored in a climatic chamber and a refrigerator keeps its open form.

Different stability results were observed for CRL-OF. A negative effect of the factors used in the climatic chamber on the stability of the immobilized CRL-OF in a dry form was noted. Residual activity results indicate low storage stability of the tested lipase. In the case of this lipase, in each study of the influence of factors in the climatic chamber (temperature, humidity, light), a decrease in the residual activity value of over 90% was obtained. This can suggest poor stability of the catalytic system to drastic storage conditions. It is difficult to unequivocally state which factor is of decisive importance. However, the designated results testify that, most likely, the storage temperature is a factor significantly affecting the decrease in lipase activity. This fact can be evidenced by the result achieved during the immobilized CRL-OF storage at 4 °C in the refrigerator. For the sample from the refrigerator, the residual activity value was 96.8 ± 5.2%, i.e., the lipase activity was very similar to that of CRL-OF tested immediately after immobilization (not stored in the refrigerator). Therefore, it was decided that the optimal temperature conditions for CRL-OF storage would be refrigerator conditions (4 °C in our research). The applied immobilization

conditions did not allow for maintaining stability during storage in the climatic chamber. It is assumed that the denaturation of catalytic proteins under the influence of temperature and related changes in the structure of the enzyme, resulting in a significant reduction of catalytic activity, are likely to occur. The created system with CRL-OF is susceptible to high temperatures. Light probably has little effect on the stability. However, this is difficult to clearly state, because, in each case, very low values of residual activity of CRL-OF were received after 7 days of storage in the climatic chamber.

Regarding CALB, the conducted storage stability studies clearly show that the created system has excellent storage stability. To our knowledge, few papers describe the use of a climatic chamber in the study of lipases [60] and other enzymes [61,62]. Our approach attempts to show that it is possible to standardize the conditions for testing enzyme systems based on the guidelines of pharmaceutical sciences. The results gained from the climatic chamber can be successfully compared between laboratories developing enzymatic catalytic systems. The study of catalytic systems (immobilized lipases) in climatic chambers is undoubtedly a novelty, insufficiently described in the literature in the field of biocatalysis. For this reason, it is important to consider the storage stage of immobilized lipases in a climatic chamber to evaluate the optimization of lipase immobilization conditions, as it was investigated and described in this work.

It should be noted that there is a noticeable relationship between the achieved results of the stability of both lipases and the effect of temperature. It is suggested that this can be related to the structure of lipases—the presence or absence of the lid. It is supposed that this structural element may significantly affect the obtained stability results. CALB with no or a small lid has high residual activity values, while CRL-OF with a lid has low storage stability. It is assumed that the structure of CRL-OF influences its increased susceptibility to high temperatures. The effect of the support on which the lipases have been immobilized is also significant. Analyzing the results, it can be seen that most likely, the immobilization conditions for CALB were more optimal and could have influenced a more optimal (proper) structural conformation on the support than in the case of CRL-OF. In addition, the support may have a much more stabilizing effect on CALB than on CRL-OF, thus enabling increased storage stability. It should be clearly emphasized that the determination of the conditions/features affecting the stability of lipases requires further research. As mentioned earlier, using climatic chambers in biocatalysis with the application of lipases is a new issue. It requires further multi-center research on the usability and unification of this method of assessing the catalytic stability of lipases.

In the literature, CALB and CRL-OF storage stability studies are described, but they are performed under different conditions, and it is difficult to compare them. Poojari et al. [63] conducted long thermal stability tests of commercially immobilized CALB (available as Novozym-435) in an organic solvent at 80 °C. They showed that the relative activity of the lipase increased with the incubation time for the first 5 days, then remained stable for the next 15 days. At the same time, free CALB activity was entirely lost within 48 h. These results show a very similar trend in stability as described in our publication. Other researchers, such as Zdarta et al. [64], studied the storage stability of CALB immobilized on silica-lignin hybrid as a novel support. The systems stored at 4 °C, after the analyzed period (20 days), retained almost all their original activity (96%), while immobilized lipase stored at 20 °C showed a decline in activity by approximately 10%. The results presented in the paper cited above can be compared to our results presented in this article regarding storing lipase in the refrigerator (4 °C) for 7 days. Regarding lipase from *Candida rugosa*, Jafarian et al. [65] studied the storage stability of immobilized *Candida rugosa* lipase on various surface-modified graphene oxide nanosheets. Thermal stability studies at 70 °C of one of the catalytic systems (CRL GON) showed a significant decrease in activity, down to 23.45% of relative activity. On the other hand, storage stability studies conducted at 25 °C showed that the residual activity of all immobilized enzymes after 50 days maintained the activity at approximately 73–79% of the initial activity. These results show a similar trend of stability as presented in our paper.

## 3. Materials and Methods

### 3.1. Materials

Octyl-Sepharose CL-4B (GE Healthcare, Uppsala, Sweden), olive oil, hydrochloric acid (concentrated), Trizma Base, and Bradford reagent were bought from Sigma-Aldrich (Steinheim, Germany). Arabic gum, phenolphthalein, citric acid monohydrate, disodium hydrogen phosphate dihydrate, monosodium hydrogen phosphate monohydrate, *o*-phosphoric acid (75%), and an analytical weighed amount of sodium hydroxide with a concentration of 0.1 M were purchased from POCH (Gliwice, Poland). Trisodium citrate was from Chempur (Piekary Śląskie, Poland). Methanol and acetone were purchased from StanLab (Lublin, Poland). Lipase B from *Candida antarctica* (CALB, produced in yeast) was from ChiralVision (Leiden, Netherlands), and lipase from *Candida rugosa* (CRL-OF) from Meito Sangyo (Tokyo, Japan). The water used in this study was filtrated by the Milli-Q Water Purification System (Millipore, Bedford, MA, USA). The storage stability of immobilized lipases was studied in a climatic chamber KBF P240 (Tuttlingen, Germany). The buffers with various pH and ionic strengths were prepared by a SevenMulti pH-meter (Mettler-Toledo, Schwerzenbach, Switzerland). The amount of immobilized enzymatic protein was determined by a UV-Vis U-1800 spectrophotometer (Hitachi, Tokyo, Japan). The lipolytic activity of lipases was evaluated with the use of a Unimax 1010 incubator (Heidolph, Schwabach, Germany) and burettes (Simax, Sázava, Czech Republic). Preparation of Octyl-Sepharose CL-4B was performed by centrifuge Eppendorf Spin Mini Plus (Hamburg, Germany) and mixer vortex Velp Scientifica ZX4 (Usmate, Italy).

### 3.2. Preparation of Octyl Sepharose CL-4B

The support suspension (110 μL) was dispensed into an Eppendorf tube. Then, 1 mL of filtered water was added to the tube with support suspension and the content was mixed using a vortex for 3 min, followed by centrifugation for 15 min at 9000 rpm. Finally, the supernatant was collected from the Eppendorf tube and the beads were weighed (50 mg).

### 3.3. Immobilization of CALB onto Octyl-Sepharose CL-4B

The immobilization procedure was performed in our laboratory based on literature data [17,32,60] with modifications. Here, 10.0 mg of CALB was placed in an Eppendorf tube (2.0 mL) with 1.0 mL of an appropriate buffer. The sample was kept for 15 min at room temperature. After this time, the lipase suspension was mixed (pipetting) and then added to the Eppendorf tube (2.0 mL) containing 50 mg of a prepared Octyl-Sepharose CL-4B support. The sample was mixed for 5 min at room temperature and kept for 14 h at 4 °C (refrigerator). After immobilization, the CALB concentration in the supernatant was determined with the use of the Bradford method. The procedures were carried out in triplicate. The process was repeated under various immobilization conditions:

a.    pH: CALB was suspended in 1.0 mL of an appropriate 100 mM buffer with pH 4, 5, and 6 (citrate buffer), 7, 8 (phosphate buffer), and 9 (Trizma Base buffer).
b.    Ionic strength: CALB was suspended in 1.0 mL of citrate buffer (pH 4) with ionic strength of: 5, 50, 100, 300, 400, 500, and 600 mM.

### 3.4. Immobilization of CRL-OF onto Octyl-Sepharose CL-4B

The immobilization procedure was performed in our laboratory based on literature data [17,18,32,60,66] with some modifications. Here, 10.0 mg of CRL-OF was placed in an Eppendorf tube (2.0 mL) with 1.0 mL of an appropriate buffer. The sample was kept for 15 min at room temperature. After this time, the lipase suspension was mixed (pipetting) and then added to the Eppendorf tube (2.0 mL) containing 50 mg of a prepared Octyl-Sepharose CL-4B support. The sample was mixed for 5 min at room temperature and allowed to stay at a temperature of 4 °C for 14 h. The procedures were carried out in triplicate. The process was repeated under various immobilization conditions:

a.   pH: CRL-OF was suspended in 1.0 mL of an appropriate 100 mM buffer with pH 4 (citrate buffer), 7 (phosphate buffer), and 9 (Trizma Base buffer).
b.   Ionic strength: CRL-OF was suspended in 1.0 mL of citrate buffer (pH 4) with ionic strength of: 50, 100, 200, 500, and 700 mM.

### 3.5. Determination of the Amount of Immobilized CALB by Bradford Method

The amount of enzymatic protein immobilized on the Octyl-Sepharose CL-4B was determined by a modified Bradford method [17,32,60,67–69]. The study was performed using the UV-Vis spectrophotometric method ($\lambda$ = 595.0 nm), measuring the absorbance of the native lipase remaining in the suspension after the immobilization process (concentration range: 1.0–10.0 mg/mL). The measurement was carried out in triplicate. The amount of CALB immobilized onto the Octyl-Sepharose CL-4B was calculated with a calibration curve equation ($R^2$ = 0.999 $\pm$ 0.001). The result was the three-sample mean. The lipase loading and protein immobilization yield (as the percentage ratio of the amount of lipase calculated as the difference between the initial amount of lipase and the amount remaining in the supernatant after immobilization onto Octyl-Sepharose CL-4B to the initial amount of lipase) [32,57,70,71] were determined based on the obtained data.

### 3.6. Lipolytic Activity of CALB and CRL-OF

The lipolytic activity of immobilized and native CALB and CRL-OF was determined by alkalimetric titration, as described in literature data [17,32,34,60,72,73]. Olive oil was used as the reaction substrate. The reaction mixture was composed of immobilized or native lipase (CALB or CRL-OF), 3.0 mL of phosphate buffer (pH 7.4, 100 mM), and 5.0 mL of emulsion containing an equal volume of olive oil and water suspension of Arabic gum (7% *w/v*). The mixture was incubated at 37 °C for 30 min at 600 rpm. Then, the reaction was discontinued by adding 5.0 mL of methanol and 5.0 mL of acetone. The titrimetric measurement was performed by applying a 0.05 M NaOH standard solution at room temperature, with phenolphthalein as the indicator. The endpoint of the titration was the change in mixture color from yellow to orange. The control was carried out without lipase (as a blank). The enzymatic activity (U, U/g support), activity recovery, relative activity, and residual activity were calculated. One unit of CALB or CRL-OF activity (U) was determined as the amount of lipase that hydrolyzed olive oil liberating 1 μmol of fatty acid per minute under the assay condition. Activity recovery (%), also defined in the literature as expressed activity [57,70,71], was described as the ratio between the activity of immobilized CALB or CRL-OF and the activity of the initial amount (10 mg) of free protein in the solution. Defining these parameters (activity recovery and expressed activity) is a matter of debate. Relative activity (%) was determined as the ratio between the activity of every sample and the maximum activity of the sample under the test conditions. Residual activity (%) was calculated as the ratio between the activity of every sample (from a climatic chamber or refrigerator) and the activity of a sample that had not been subjected to a storage stability test in a climatic chamber or refrigerator. Analyses were performed in triplicate.

### 3.7. Climatic Chamber Storage Stability Tests of CALB and CRL-OF in Dry Form

The procedure for the storage stability studies of immobilized CALB and CRL-OF was prepared in our laboratory. After immobilization in citrate buffer (pH 4, 500 mM) followed by drying at room temperature (3 days), the Octyl-Sepharose CL-4B beads with enzymes were stored in a climatic chamber KBF P240. The temperature was maintained at 65 °C, the humidity was 75%, and the visible spectral range was 400–800 nm. The studies were conducted for 7 days. The samples were stored in various vials: open glass, closed glass, dark glass open, and dark glass closed. The immobilized lipase beads after drying were also stored in a refrigerator. After 7 days, the enzymatic activity (U and U/g support) and residual activity (%) of immobilized lipases were evaluated by the method described in Section 3.6. The storage conditions of immobilized lipases, in dry form, are shown in Table 3.

**Table 3.** The storage conditions of immobilized CALB and CRL-OF in dry form in the climatic chamber and in the refrigerator.

| Dry Form of Lipases | | | | |
|---|---|---|---|---|
| Type of Glass | Type of Tube | Incubation Time (days) | Temperature (°C)/Humidity (%) | Vis (400–800 nm) |
| Dark | Open [1] | | 65/75 | - |
| | Closed [1] | | 65/- | |
| Transparent | Open [1] | 7 | 65/75 | + |
| | Closed [1] | | 65/- | |
| | Closed [2] | | 4/- | - |

[1] The conditions in the climatic chamber KBF P240. [2] The conditions in the refrigerator.

## 4. Conclusions

Immobilization of CALB on Octyl-Sepharose CL-4B support using the developed protocol: citrate buffer at pH 4 and ionic strength of 500 mM, seems to be the optimal approach to obtain a biocatalyst with high lipolytic activity—hyperactivation (recovery activity $116.10 \pm 1.70\%$) in the tested reaction system (olive oil as a substrate). The immobilized CALB, subjected to drastic temperature and humidity conditions in a climatic chamber, was characterized by good storage stability (residual activity $218 \pm 7.3\%$ of dry form after 7 days). Lower stability results were achieved for the immobilized CRL-OF. To our knowledge, the use of a climatic chamber to study the storage stability of immobilized lipases, as described in this paper, is a new solution not reported in the literature. The climatic chambers guarantee that storage testing is carried out in accordance with the International Conference on Harmonization (ICH) guidelines. Importantly, the test conditions in the chamber are stable and repeatable and meet the stringent requirements of the pharmaceutical industry. The proposed storage method has the potential for standardization and unification of the inter-laboratory evaluation of the created lipase-biocatalysts, which is extremely important from the point of view of their commercial application.

**Author Contributions:** Conceptualization, T.S.; methodology, T.S.; formal analysis, T.S. and J.D.; investigation, T.S., J.S., R.M., N.K. and J.D.; resources, T.S., J.S. and M.P.M.; writing—original draft preparation, T.S. and J.S.; writing—review and editing, R.M., N.K., J.D. and G.G.H.; visualization, T.S. and J.D.; supervision, G.G.H., D.W.-Ś. and M.P.M.; project administration, T.S.; funding acquisition, T.S. and J.S. All authors have read and agreed to the published version of the manuscript.

**Funding:** This work was partially supported by: Excellence Initiative - Debuts, under the "Excellence Initiative—Research University" program, NCU in Toruń - 9/2022/Debiuty3 and 6/2022/Debiuty3, and the "Excellence Initiative—Research University" program, NCU in Toruń, 118/2021/Grants4NCUStudents.

**Data Availability Statement:** Not applicable.

**Acknowledgments:** The authors wish to express their sincere thanks to Meito Sangyo Co. (Japan) for the supply of lipase OF.

**Conflicts of Interest:** The authors have no conflict of interest to declare.

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
