# Peer review of "Climatic Chamber Stability Tests of Lipase-Catalytic Octyl-Sepharose Systems"

_catalysts, doi:10.3390/catal13030501_

Round 1

Reviewer 1 Report

Manuscript

Climatic chamber stability tests of catalytic Octyl-Sepharose  systems: lipases immobilization protocols

 Tomasz Siódmiak, Joanna Siódmiak, RafaÅ‚ Mastalerz, Natalia Kocot, Jacek DulÄ™ba, Gudmundur G. Haralds- son, Dorota WÄ…tróbska-Åšwietlikowska, MichaÅ‚ Piotr Marszałł

This research is a very innovative and encompasses many tasks associated with enzymology science and development of biocatalysts for industrial biotechnology. In this research the storage stability of immobilized lipases was characterized in the climatic chambers and results could be applied in chemical, pharmaceutical, and biotechnological industries. This manuscript revealed the developed protocols for immobilization via interfacial activation of lipase B from Candida antarctica (CALB) and lipase OF from Candida rugosa (CRL-OF) on the Octyl-Sepharose CL-4B support. The described method has the potential for standardization and unification of the inter-laboratory evaluation of the created lipase-biocatalysts, which could be very significant in the chemical, pharmaceutical, and biotechnological industries.

Some comments and suggestions to the authors:

       1) It should be written „pH of the reaction medium for CALB is 7.4“ in the paragraph 69.

2)     It should be written correctly, „Important, yeast Candida rugosa in the paragraph 81.

3)     In the Figure 2, in the y axis should be written only one symbol %.

4)     In the Figure 4, in the y axis should be written only one symbol %.

5)     It should be written „ionic strength - 0.5 M“ in the paragraph 238;

6)     It should be written 1 M in the paragraph 243.

7)     It should be written 500 mM in the paragraph 265.

8)     It should be written 500 mM in the paragraph 330.

9)     It should be written the sentence more clearly „2.0 M KCl.? De Melo et al. [58] described the immobilization of CRL on Octyl-Sepharose in a buffer“ in the paragraph 356.

10) It should be written 500 mM in the paragraph 570.

11) To check the whole list of references according to the journal „Catalysts“ requirements. 

i

Author Response

The answer for Reviewer 1

I would like to thank you very much for thoroughly reading the manuscript, issuing an opinion, and presenting valuable insights on quality. The suggested changes were included in the revision. Please see the revised version.

       1) It should be written „pH of the reaction medium for CALB is 7.4“ in the paragraph 69.

Thank you for your suggestion. The change has been introduced.

2)     It should be written correctly, „Important, yeast Candida rugosa in the paragraph 81.

Thank you for your suggestion. The change has been introduced.

3)     In the Figure 2, in the y axis should be written only one symbol %.

Thank you for your suggestion. The change has been introduced.

4)     In the Figure 4, in the y axis should be written only one symbol %.

Thank you for your suggestion. The change has been introduced.

5)     It should be written „ionic strength - 0.5 M“ in the paragraph 238;

Thank you for your suggestion. The change has been introduced.

6)     It should be written 1 M in the paragraph 243.

Thank you for your suggestion. The change has been introduced.

7)     It should be written 500 mM in the paragraph 265.

Thank you for your suggestion. The change has been introduced.

8)     It should be written 500 mM in the paragraph 330.

Thank you for your suggestion. The change has been introduced.

9)     It should be written the sentence more clearly „2.0 M KCl.? De Melo et al. [58] described the immobilization of CRL on Octyl-Sepharose in a buffer“ in the paragraph 356.

Thank you for your suggestion. The change has been introduced.

10) It should be written 500 mM in the paragraph 570.

Thank you for your suggestion. The change has been introduced.

11) To check the whole list of references according to the journal „Catalysts“ requirements. 

Thank you for your suggestion. The whole list of references has been checked. 

Reviewer 2 Report

Title: Climatic chamber stability tests of catalytic Octyl-Sepharose systems: lipases immobilization protocols

The novelty and the quality of the manuscript are good and it does not need extensive improvement before publication. It is carefully organized and written. It is easy to follow it and contains clear comments and conclusions. 

In my opinion, this manuscript is very detailed and meticulous, it covers all the literature in the field with critical point of view. The topic have been completely covered and is well connected through the text. There is a significant  novelty in presented topic.  For all these reasons, I can only recommend the acception of the manuscript after minor revision. I only have one minor suggestion.

The superiority of  the use of a climatic chamber to test the storage stability of a dry form of the studied lipases immobilized on Octyl-Sepharose CL- 4B  than other used methods should be more emphasized.

Author Response

The answer for Reviewer 2

I would like to thank you very much for thoroughly reading the manuscript, issuing an opinion, and presenting valuable insights on quality. The suggested changes were included in the revision.

Comments: The novelty and the quality of the manuscript are good and it does not need extensive improvement before publication. It is carefully organized and written. It is easy to follow it and contains clear comments and conclusions. 

In my opinion, this manuscript is very detailed and meticulous, it covers all the literature in the field with critical point of view. The topic have been completely covered and is well connected through the text. There is a significant  novelty in presented topic.  For all these reasons, I can only recommend the acception of the manuscript after minor revision. I only have one minor suggestion.

The superiority of  the use of a climatic chamber to test the storage stability of a dry form of the studied lipases immobilized on Octyl-Sepharose CL- 4B  than other used methods should be more emphasized.

Response: Thank you for your suggestions.

The superiority of testing in a climatic chamber over testing using other methods/techniques lies in the fact that the climatic chambers guarantee that storage testing is carried out in accordance with International Conference on Harmonization (ICH) guidelines. Importantly, the test conditions in the chamber are stable and repeatable and meet the stringent requirements of the pharmaceutical industry. The proposed storage method has the potential for standardization and unification of the inter-laboratory evaluation of the created lipase-biocatalysts, which is extremely important from the point of view of their commercial application.

This information was included in Section 4. - Conclusions

Reviewer 3 Report

According to the title of the paper, the main content should be the study of lipase immobilization conditions on its stability in the climatic chamber, but main part of the paper is devoted to study the effects of pH and ionic strength on the activity recovery of immobilized lipase, rather than the stability of the immobilized lipase in the climate chamber, and whether the main content of the study deviates from the title.

In this paper, the introduction is comprehensive, the method is clear, and the results and discussion also very thorough, but it seems that too many words are written, writing of the paper is not succinct enough.

The percent (%) in Figure 2 and Figure 4 should be placed behind the title of the vertical coordinates, just as in Figure 1.

Sorry for my less knowledge about the climatic chamber, the above opinion may not be correct.

Author Response

The answer for Reviewer 3

I would like to thank you very much for thoroughly reading the manuscript, issuing an opinion, and presenting valuable insights on quality. The suggested changes were included in the revision.

Comments: According to the title of the paper, the main content should be the study of lipase immobilization conditions on its stability in the climatic chamber, but main part of the paper is devoted to study the effects of pH and ionic strength on the activity recovery of immobilized lipase, rather than the stability of the immobilized lipase in the climate chamber, and whether the main content of the study deviates from the title.

Response: Thank you for your suggestions.

The first part of the paper describes the effect of pH and ionic strength of buffers used during immobilization on lipase activity in order to select systems with the highest catalytic activity. The activity was expressed as activity recovery as well as relative activity in order to clearly and transparently present the results and allow to selection of the immobilization conditions in which the catalyst is characterized by the highest activity. This study phase was necessary for the second stage (stability study) to be carried out. In the second stage, stability studies of the biocatalyst with the highest catalytic parameters were performed. At this phase, the stability of the system to high temperature, light, and humidity was assessed. The evaluation of the stability was carried out by determining the residual activity parameter, based on the "U" activity value, i.e. the activity of lipase after 7 days of storage and lipase immediately after immobilization was compared. It should be emphasized that it was decided at the study design stage that the stability tests would be performed for catalysts with the highest activity. The authors would like to draw attention to the different stability of the CALB and CRL-OF lipases and the need to optimize the immobilization procedures adapted to the specific lipase, as well as the importance of the high ionic strength of using the citrate buffer.

We understand your opinion but it is very difficult to choose the best catalyst without the immobilization optimization step. The pH value and ionic strength of the buffers used have a crucial impact on enhancing the activity of immobilized lipase therefore they have been extensively described. The presented results are the first part of the project, and the next papers describing the use of a climatic chamber in the study of stability are in preparation. This is the first paper in the literature describing the application of the climatic chamber in this type of immobilized lipases study. Due to the important aspect of the application of a climatic chamber in lipase storage studies, the title of the paper was proposed in this form. The title of the paper has been slightly modified.

“Climatic chamber stability tests of lipase-catalytic Octyl-Sepharose systems”

Comments:  In this paper, the introduction is comprehensive, the method is clear, and the results and discussion also very thorough, but it seems that too many words are written, writing of the paper is not succinct enough.

Response: We agree that it is possible to write in a more succinct way the paper, however, the conducted studies, due to their innovative potential, in our opinion, require extensive commentary and analysis of literature data, hence the comprehensive descriptions in the discussion. The authors attempt to draw attention to the very different stability testing techniques described in the literature and the need to standardize them.

Comments: The percent (%) in Figure 2 and Figure 4 should be placed behind the title of the vertical coordinates, just as in Figure 1.

Response: Thank you for your suggestion. The changes have been introduced.